# Empirical Analysis of Preferences of Older Adults for Care Facilities in Japan: Focusing on Household Structure and Economic Status

**DOI:** 10.3390/healthcare11131843

**Published:** 2023-06-25

**Authors:** Jinhan Wang, Ziyan Wang, Bing Niu

**Affiliations:** Graduate School of Economics, Osaka Metropolitan University, Osaka 599-8531, Japan; sn22770g@st.omu.ac.jp (Z.W.); niubing@omu.ac.jp (B.N.)

**Keywords:** older adults, household structure, economic status, preferences, care facilities

## Abstract

Japan is advancing into a super-aged society at an unprecedented speed, and the proportion of the elderly population will continue to rise. The number of older adults needing nursing care will also increase with the aging population. We used a cross-sectional dataset of older Japanese adults to examine their future preferences for care facilities and their relationship with individual characteristics, household structure, and economic status. We further focused on a subgroup of those who lived alone and were experiencing poverty and examined their care needs through their choice of care facilities. We found the following results from multinomial logit and probit regressions. First, compared with living alone, older adults who live with their spouses or other members prefer to live in their own houses. Second, older adults experiencing poverty preferred to choose facilities geared towards low-income groups, while wealthy older adults preferred to choose fee-based nursing homes/nursing homes with diverse services and high costs. Third, single older adults in poverty were less likely to choose to live in facilities. Covariates such as sex, age, and health status also mattered in their choices. Therefore, tailoring the formal care services to the preferences and actual needs of older adults is imperative.

## 1. Introduction

In Japan, the demographic structure has changed over the past century. The proportion of the young population has declined consistently, while that of the elderly population continues to grow, exceeding 21% in 2010 and reaching 29% by 2020 (Figure 1) [1]. Soon, Japan will face a “population and social security problem”, where the population aged 75 and over is expected to grow to 21.5 million, approximately one in five people of the total population, by 2025. By 2040, it is predicted that 34.8% of the population will be aged 65 years or older, and 19.7% will be aged 75 years or older [2].

In anticipation of an aging population with an increasing number of older adults needing nursing care, the long-term care insurance system (LTCI) was introduced in 2000 [3]. It aims to maintain dignity and an independent daily life routine according to each person’s ability level for those needing long-term care [4]. It was also created to allow users to select comprehensive nursing care and health services from a wide range of services. Figure 2 shows the changing trend of nursing staff and people certified as needing long-term care. Over the past 20 years, users and nursing staff have increased. Figure 3 shows the change in the capacity of care facilities according to facility type for older adults. The capacity of “fee-based nursing homes” is increasing dramatically. In contrast, the capacity of “low-cost nursing homes” and “nursing homes for older adults”, aimed at older adults experiencing poverty, has not changed significantly.

Although the demand and supply of formal care services are increasing, Japan faces two problems with the current system. First, the LTCI lacks financial resources [9]. Fifty percent of the LTCI’s financial resources are public funds and taxes, and the remaining fifty percent are insurance premiums paid by insured persons aged 40 years and older [4,10]. With the aging population, the need for nursing care is increasing, and the cost of nursing care is also increasing correspondingly. In 2019, the cost of long-term care insurance reached 10 trillion JPY, about three times that in 2000, when the long-term care insurance system was established [11]. By 2025, when the baby boomer generation reaches the age of 75 or older, the need for nursing care will continue to rise, and care costs are expected to increase further [2]. From 2025 onwards, the population of insured persons aged 40 years and over who bear long-term care insurance premiums will decrease, and the government is facing an important issue in securing financial resources.

Second, there must be more human resources in Japan’s nursing industry [9]. To secure the quantity of quality nursing staff, Japan has made efforts to promote participation, improve qualifications, and improve working environments and treatment [12]. Although the number of nursing staff is increasing, as shown in Figure 2, it still cannot keep up with the growth of the older adult population and is in a complete shortage [9]. With limited financial and human resources, policies need to be put forward to better allocate nursing resources so that the formal care provided meets the actual needs of older adults so that they can live healthier lives. Recent policies have focused on meeting the preferences and choices of care for older adults [13]. In 2017, the World Health Organization (WHO) guidelines on Integrated Care for Older People (ICOPE) proposed evidence-based recommendations that each country should place the preferences and needs of older adults at the center to coordinate care efficiently [14].

Previous studies in Japan on the care preferences of older adults have focused on their relationship with social networks [15]; age, period, and cohort (A-P-C) [16]; and sex, cohabitation, subjective health status, and family savings [17]. Kikuchi (2014) used an individual national dataset to examine the effects of social networks on the preference for long-term care among older adults and found that older adults with wide social networks were less likely to choose care facilities [15]. Sugisawa et al. (2019) used a repeated cross-sectional dataset to examine differences in preferences for long-term care according to A-P-C. They also analyzed the interaction effects of A-P-C with sex, family structure, and activities of daily living on preferences. They found that younger participants were likelier to prefer community and institutional care services to informal care. The age effect was stronger in women and in respondents who lived alone [16]. Zhang et al. (2023) used cross-sectional data on older adults to examine the factors affecting their preference for long-term care by focusing on sex, age, cohabitation, subjective health status, and family savings [17].

The main objective of this study was to shed light on the preferred residential and care arrangements in the later life of older adults. We used a cross-sectional dataset of Japanese older adults aged 65 and over in Sakai City to examine their preferences for care facilities in the future and their relationship with personal characteristics (age, gender, instrumental activities of daily living (IADL), and subjective health), with a particular focus on household structure and economic status. We then focused on older adults who lived alone and were experiencing poverty to reveal their care needs through their choice of care facilities.

In 2019, the relative poverty rate for “single households aged 65 and over” in Japan was 29.9% [18], which means that nearly one-third of older adults living alone are in poverty. The number of older adults experiencing poverty living alone is predicted to increase rapidly, owing to an increase in unmarried and divorced populations [19]. It has also been reported that the utilization of healthcare services varies by income level among older adults. Compared with middle- and high-income groups, older adults with low incomes used fewer healthcare services, such as inpatient care and LTCI home care, and the financial burden of healthcare expenditure for the low-income group was higher than that of middle- and high-income groups [20]. Owing to the limited number of empirical studies targeting older adults experiencing poverty in single households, we focused on this subgroup of people. We clarified their needs for each type of care facility.

## 2. Materials and Methods

### 2.1. Data

We applied a dataset from a 2019 survey conducted in Sakai City regarding the living conditions, health, and welfare of the older adults (aged 65 years and over) in Osaka Prefecture, Japan [21]. A sample of 9400 respondents who had not been certified by LTCI and those who had been certified with the mildest levels of needing support, “Requiring help 1–2”, were randomly selected for the survey. The Health and Welfare Bureau of Sakai City, Japan, granted permission to use the data after applying for information disclosure.

The target sample in our analysis were those who had not been certified by LTCI and were currently living in their own homes and those who answered the question by themselves without any assistance from family members/relatives and others present. We included a sample size of 5332 older adults in our analysis.

There are several advantages to using this dataset in our study. First, it is a unique dataset that encompasses older adults’ various characteristics and clear living conditions. It contains age, sex, household structure, economic status, utilization, and preference for long-term care services. It also includes information on instrumental activities of daily living (IADL) and the subjective health status of older adults. We included these variables in the analysis because changes in IADL and subjective health have been found to be significant predictors of changes in preferences among older adults [22].

Second, this dataset was sufficiently representative, as the sex ratio and age-specific population structure of Sakai City were consistent with Japan’s at the time of this study. A comparison of these characteristics between Sakai City and the entire country of Japan is shown in Appendix A.

### 2.2. Dependent Variables

The dependent variables in Table 1 were generated based on the answers to the following questions in the questionnaire [21], “What kind of facilities would you like to live in in the future?”.

“I want to continue living in my current house (LIH).”“I want to live in an LTCI care facility for people certified as requiring long-term care, such as a special nursing home for older adults (LCIF).”“I want to live in a relatively small-scale special nursing home for older adults or a group home for people with dementia (SNH).”“I want to live in a welfare facility for low-income older adults, such as a low-cost nursing home (care house), etc. (LWF).”“I want to live in a fee-based nursing home or nursing home with diverse services and high cost for the older adults (FSH).”“Other, such as living with relatives, etc. (OTHS).”

### 2.3. Independent Variables

The leading independent variables were household structure (living alone/couple only/other structures), economic status (experiencing poverty/normal/wealthy), and an interaction term of “Living Alone and Experiencing Poverty”, indicating the target subgroup of older adults experiencing poverty who are living alone. Economic status was categorized based on the self-assessed answers of the respondents, based on the question, “How do you think about your economic status based on your current living situation?”.

We hypothesized that future care preferences of older adults would change because of differences in their household structures. Living with other family members makes them more likely to become caregivers when they need care [23]. Moreover, family relationships and the level of support available from the family are important determinants of preferences for care, and the opinions and views of family members regarding care also affect older adults’ preferences [24].

Moreover, household economic resources are also significant predictors of preferences: families with higher incomes could facilitate the purchase of home-based assistance and might allow a change of dwelling or move towards residential care [25].

We also used sex, age (ranges 65–69, 70–74, 75–79, and ≥80), IADL (dependent/independent), and subjective health status (healthy/unhealthy) as covariates. IADL was categorized based on five questions: “Can you take the bus and tram alone?”; “Can you buy food and daily necessities alone?”; “Can you cook by yourself?”; “Can you pay bills by yourself?”; “Can you deposit and withdraw money by yourself?”. If a respondent answered “No” to at least one of these questions, the respondent was categorized as “dependent”.

### 2.4. Estimation Framework

We conducted two analyses to examine the relationship between older adults’ preferences for care facilities and independent variables, focusing on household structure and economic status.

In Analysis I, we performed a multinomial logit regression [26] to examine the choice between different care facilities, as shown in Figure 4.

The estimation model is shown in Equation (1).
(1)$$P_{ij}=\mathrm{Pr}\left[\left.preference_{i}=j\right|X_{i}\right]=\frac{\exp\;\left(X_{i}^{’}\beta_{j}\right)}{\sum_{k=1}^{6}\exp\;\left(X_{i}^{’}\beta_{k}\right)}=\frac{\exp\;\left(H_{i}\alpha_{j}+E_{i}\beta_{j}+x_{i}\gamma_{j}+\mu \right)}{\sum_{k=1}^{6}\exp\;\left(H_{i}\alpha_{k}+E_{i}\beta_{k}+x_{i}\gamma_{k}+\mu \right)},\;j=1,\ldots ,\;6.$$

The dependent variable $P_{ij}$ represents the probability of older adults *i* choosing care facility *j* from among all six categories. $H_{i}$ is the household structure of older adult *i*. $E_{i}$ is the economic status of older adult *i*. $x_{i}$ represents the other covariates.

In Analysis II, we set an interaction term “Living Alone and Experiencing Poverty” to represent those older adults experiencing poverty in single households, and performed a probit regression [26] to examine their preferences with respect to each type of care facility, as shown in Figure 5.

The estimation model is shown in Equation (2).
(2)$$P_{i}=\mathrm{Pr}\left[\left.preference_{i}=1\right|X_{i}\right]=F\left(X_{i}^{’}\beta \right)=\Phi \left[H_{i}\alpha +E_{i}\beta +\left(H_{i}\times E_{i}\right)\delta +x_{i}\gamma +\mu \right]$$

The dependent variable $P_{i}$ represents the probability of older adults *i* choosing a particular type of care facility. $H_{i}$ is the household structure (dummy variable) of older adult *i*. $E_{i}$ is the economic status (dummy variable) of older adult *i*. $H_{i}\times E_{i}$ is the interaction term “Living Alone and Experiencing Poverty”. $x_{i}$ represents the other covariates.

We calculated the marginal effect $\left(\partial P_{i}/\partial X_{ik}\right)$ of change in a regressor $X_{ik}$ on the conditional probability that $preference_{i}=1\;$from Equation (2), shown as,
(3)$$\frac{\partial \mathrm{Pr}[preference_{i}=1|X_{i}]}{\partial X_{ik}}=F^{\prime }\left(X_{i}^{’}\beta \right)\beta_{k}=\phi \;\left(X_{i}^{’}\beta \right)\beta_{k}$$

## 3. Results

### 3.1. Estimation Results of Analysis I

The odds ratios of household structure, economic status, and other covariates on older adults’ preference for care facilities, estimated from the multinomial logit models, are summarized in Table 2. The reference group was those who choose to live in their own houses (LIH).

Regarding the household structure, we found that compared with those living alone, older adults who live with their spouses or other members prefer to live in their own houses (LIH) rather than different types of facilities, such as a care facility of the LTCI (LCIF) (couple only: 0.61, *p* < 0.05; other structures: 0.29, *p* < 0.01), a welfare facility for low-income people (LWF) (couple only: 0.51, *p* < 0.01; other structures: 0.64, *p* < 0.01), a fee-based nursing home/a nursing home with diverse services and a high cost (FSH) (couple only: 0.52, *p* < 0.01; other structures: 0.46, *p* < 0.01), and other situations (OTHS) (couple only: 0.62, *p* < 0.1).

Older adults with a wealthy economic status were 1.89 times more likely to choose fee-based nursing homes/nursing homes with diverse services and high costs (FSH) (OR: 1.89, *p* < 0.01); however, they were less likely to choose a care facility of the LTCI (LCIF) (OR: 0.50, *p* < 0.05) or a welfare facility for low-income people (LWF) (OR: 0.26, *p* < 0.01). Older adults experiencing poverty were 2.37 times more likely to choose low-cost nursing homes (LWF) (OR: 2.37, *p* < 0.01) and less likely to choose fee-based nursing homes/nursing homes with diverse services and high costs (FSH) (OR: 0.54, *p* < 0.01).

Other covariates, such as sex, age, and health status, also significantly affected their choices. Compared with the choice of living at home, men were less likely to choose welfare facilities for low-income people (LWF) (OR: 0.66, *p* < 0.01), and those aged *<* 74 years preferred other situations (OTHS) (aged 65–69 years: 2.34, *p* < 0.05; aged 70–74 years: 2.07, *p* < 0.05). Older adults with low levels of IADL were more likely to choose to live in their own houses than in other situations (OTHS: 0.55, *p* < 0.05). Healthy older adults preferred to live in their own houses compared with those living in care facilities, such as LCIFs (OR: 0.56, *p* < 0.05), SNHs (OR: 0.48, *p* < 0.01), LWFs (OR: 0.63, *p* < 0.01), and OTHS (OR: 0.62, *p* < 0.1).

### 3.2. Estimation Results of Analysis II

The marginal effects of household structure, economic status, their interaction term, and other covariates on older adults’ preferences for each care facility estimated from the probit models are summarized in Table 3.

Compared with other types of household structures, older adults who lived alone were less likely to choose to live in their own houses (LIH) by 12.8%. They were more likely to choose to live in care facilities such as care facilities of the LTCI (LCIF) by 2.0%, welfare facilities for low-income people (LWF) by 5.9%, and fee-based nursing homes/nursing homes with diverse services and high costs (FSH) by 3.8%.

Older adults experiencing poverty were also less likely to choose to live in their own houses (LIH) by 7.9% or fee-based nursing homes/nursing homes with diverse services and high costs (FSH) by 3.0%, but were more likely to choose welfare facilities for low-income people (LWF) by 9.1%.

The focused subgroup of older adults experiencing poverty who lived alone was less likely to choose welfare facilities for low-income people (LWF) or fee-based nursing homes/nursing homes with diverse services and high costs (FSH) by 2.7% and 1.9%, respectively, but was more likely to choose other situations (OTHS) by 3.4%. Regarding other covariates, men and healthy older adults were 4.8% and 9.7% more likely to choose to live at home, respectively. Respondents under 74 were more likely to choose other situations (OTHS) by 2.1–2.6%.

## 4. Discussion

Based on the results of this study, several observations emerged. First, regarding the structure of households, older adults who live with their spouses or other members prefer to live in their own houses (LIH) in the future compared with those living alone. In 2019, the number of people certified as needing long-term care or support reached 6.67 million in Japan [6] and is expected to continue to rise [27]. A national survey in Japan conducted in 2019, the Comprehensive Survey of Living Conditions, reported that 68% of people in need of care are cared for by family members [23]. Family caregivers are important in supporting older family members to continue living at home and delaying admission to care facilities [28,29]. However, family caregiving has adverse impacts on caregivers in terms of their social interactions [30], quality of life [31,32], and mental health [33]. It has been reported that 74% of people want to receive long-term care at home when a need arises [34], and supporting informal care and their caregivers is becoming an important policy issue [35].

Second, regarding economic status, older adults experiencing poverty prefer to choose facilities for the low-income group (LWF). In contrast, wealthy older adults prefer to choose fee-based nursing homes/nursing homes with diverse services and high costs (FSH), which cost more and provide diverse services according to actual needs. Based on the Gini Index 2019, income disparity among older adults was higher than among other age groups [36,37]. Differences in economic resources were found to be a significant predictor of changes in preferences between wealthy older adults and low incomes [25], which might eventually affect the health and well-being of older adults.

Third, regarding the focus group of single older adults in poverty, they were less likely to choose to live in facilities, either facilities for low-income people (LWF) or fee-based nursing homes/nursing homes with diverse services and high costs (FSH), but rather chose other situations (OTHS). As the number of older adults living alone increases rapidly, lonely deaths have become a major social issue [38]. Increasing access to formal care among the target group (i.e., older adults experiencing poverty in single households) is an important policy issue. In Japan, using formal care services based on the long-term care insurance system is complicated and sometimes difficult for older adults to understand [39]. Nearly 35% of older care recipients have inadequate care literacy (the ability to obtain basic knowledge about the LTCI system and caregiving) [40]. Improving the care literacy of older adults is important for better use of formal care services.

## 5. Conclusions

In this study, we used a cross-sectional dataset of older Japanese adults to examine their future preferences for care facilities and their relationship with individual characteristics, focusing on household structure and economic status. We further focused on a subgroup of those who lived alone and were experiencing poverty and examined their care needs through their choice of care facilities. We found the following results: first, compared with those living alone, older adults who live with their spouses or other members prefer to live in their own houses in the future; second, older adults experiencing poverty prefer to choose facilities geared towards low-income groups, while wealthy older adults prefer to choose fee-based nursing homes/nursing homes with diverse services and high costs; third, single older adults in poverty are less likely to choose to live in facilities. Covariates such as sex, age, and health status also mattered in their choices.

This study had several limitations that should be acknowledged and addressed in future studies. First, the data used in this study were cross-sectional, which limited the causal inference of the relationship between care preferences and independent variables. Longitudinal data are needed to examine the dynamic changes in preferences and other variables over time. Second, in addition to household structure and economic status, other factors can be considered predictors of preferences. As mentioned in the Discussion, the utilization of formal care services is based on the LTCI system, which is complicated and difficult for older adults and family members to understand [39]. Older adults with inadequate care literacy may not be able to easily express and request the services they want or understand whether the presented care services meet their actual needs. Therefore, care literacy might be an important predictor of preference and choice and needs to be examined in future studies.

The aging of the Japanese population will continue to progress significantly. It is important to solve the major problems that arise in the supply of long-term care services: the lack of financial resources and labor supply. It is also imperative that formal care services be tailored to the preferences and actual needs of older adults who need care in the future.

## Figures and Tables

**Figure 1 healthcare-11-01843-f001:**
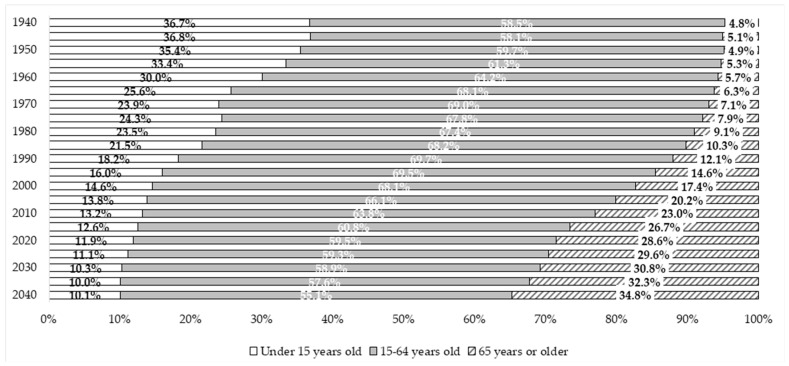
Changes in population ratio by age group in Japan (as of October 1st of each year) (1940–2040). Data source: 1940–2020: Ministry of Internal Affairs and Communications, Population Estimates [1]. 2025–2040: National Institute of Population and Social Security Research (IPSS), Population projections for Japan [2].

**Figure 2 healthcare-11-01843-f002:**
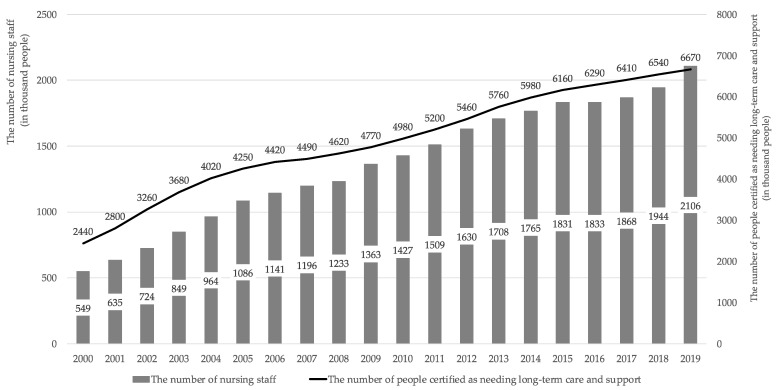
The number of nursing staff and people certified as needing long-term care and support (2000–2019). Data source: Ministry of Health, Labor and Welfare [5,6]. Notes: “The number of nursing staff” is the number of staff engaged in long-term care service establishments and long-term care insurance facilities eligible for long-term care insurance benefits.

**Figure 3 healthcare-11-01843-f003:**
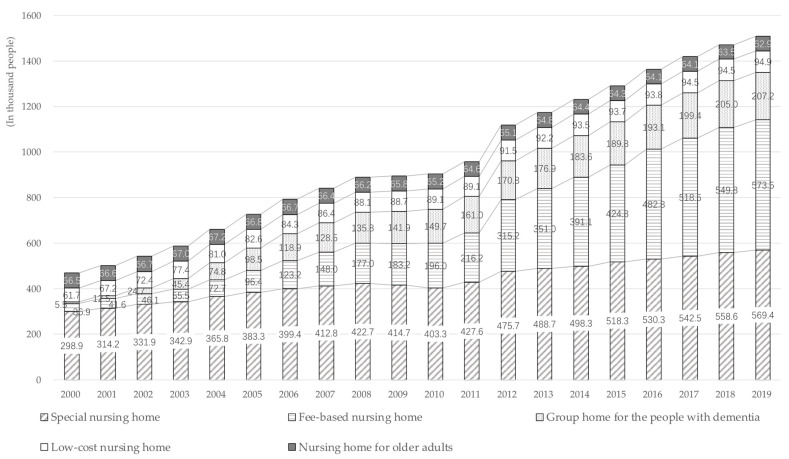
The maximum capacity of care facilities by facility type (2000–2019). Data source: Ministry of Health, Labor and Welfare (2000–2004) [7]; Cabinet Office (2005–2019) [8]. Notes: “Special nursing home”: facility for older adults aged 65 and over who are certified as needing long-term care at certified level 3 and above; “Fee-based nursing home”: care facility with a high cost; “Group home for people with dementia”: facility for older adults aged 65 and over who are diagnosed with dementia and are certified at level 1 and above, and those who live in the city where the facility is located; “Low-cost nursing home”: facility for older adults experiencing poverty who are not able to live independently; “Nursing home for older adults”: facility for older adults experiencing poverty aged 65 and over who are not certified as needing long-term care and those who can live independently.

**Figure 4 healthcare-11-01843-f004:**
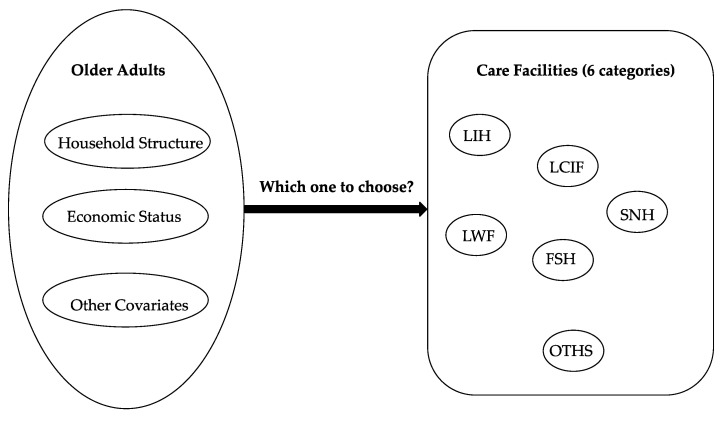
The framework of Analysis I.

**Figure 5 healthcare-11-01843-f005:**
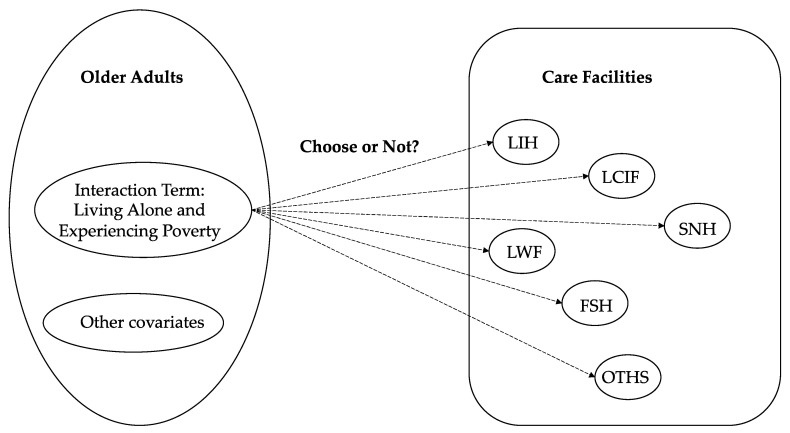
The framework of Analysis II.

**Table 1 healthcare-11-01843-t001:** Descriptive statistics.

Variable	Obs	Mean	Std. Dev.	Min	Max
Dependent Variables					
Analysis I: Care facilities (Categories of care facilities)	5178	0.72	1.41	0	5
Analysis II: Each care facility					
(1) LIH	5332	0.74	0.44	0	1
(2) LCIF	5332	0.03	0.18	0	1
(3) SNH	5332	0.03	0.18	0	1
(4) LWF	5332	0.09	0.28	0	1
(5) FSH	5332	0.05	0.22	0	1
(6) OTHS	5332	0.03	0.17	0	1
Main Independent Variables					
Household structure:					
Living alone	5226	0.21	0.41	0	1
Couple only	5226	0.49	0.50	0	1
Other structures	5226	0.30	0.46	0	1
Economic status (self-assessed):					
Experiencing poverty	5308	0.34	0.47	0	1
Normal	5308	0.58	0.49	0	1
Wealthy	5308	0.08	0.27	0	1
Interaction term:					
Living Alone and Experiencing Poverty	5205	0.08	0.28	0	1
Other Covariates					
Sex:					
Male	5332	0.43	0.49	0	1
Age:					
65–69 y	5332	0.23	0.42	0	1
70–74 y	5332	0.28	0.45	0	1
75–79 y	5332	0.27	0.45	0	1
80+ y	5332	0.22	0.41	0	1
Health status:					
IADL (dependent)	5216	0.11	0.31	0	1
Subjective health (healthy)	5257	0.80	0.40	0	1

Data source: A Survey on Older Adults of Sakai City (2019) [21].

**Table 2 healthcare-11-01843-t002:** Odds ratios of household structure and economic status and other covariates on preference for care facilities estimated from multinomial logit models.

Variable	LCIF (ref: LIH)			SNH (ref: LIH)			LWF (ref: LIH)			FSH (ref: LIH)			OTHS (ref: LIH)		
	OR		SE	OR		SE	OR		SE	OR		SE	OR		SE
Household structure (ref: living alone)															
Couple only	0.61	**	0.11	1.08		0.23	0.51	***	0.07	0.52	***	0.08	0.62	*	0.14
Others	0.29	***	0.07	0.80		0.19	0.64	***	0.09	0.46	***	0.08	0.81		0.19
Economic status (self-assessed) (ref: normal)															
Experiencing poverty	1.17		0.20	1.20		0.21	2.37	***	0.25	0.54	***	0.09	1.13		0.21
Wealthy	0.50	**	0.20	0.67		0.24	0.26	***	0.10	1.89	***	0.34	1.33		0.37
Sex (ref: female)															
Male	0.88		0.15	0.83		0.14	0.66	***	0.07	0.79		0.11	1.02		0.18
Age (ref: 80+ y)															
65–69 y	0.95		0.25	1.17		0.28	1.17		0.19	1.25		0.24	2.34	**	0.66
70–74 y	1.22		0.28	1.04		0.24	1.15		0.17	0.87		0.17	2.07	**	0.57
75–79 y	1.34		0.31	0.88		0.21	1.07		0.16	1.14		0.21	1.40		0.42
Health status															
IADL (dependent)	0.83		0.23	0.90		0.24	0.91		0.16	1.02		0.24	0.55	**	0.20
Subjective health (healthy)	0.56	**	0.10	0.48	***	0.09	0.63	***	0.08	0.86		0.15	0.62	*	0.13
Constant	0.11	***	0.03	0.08	***	0.02	0.21	***	0.04	0.15	***	0.03	0.04	***	0.01
Number of obs	4906														

Notes: “LIH”: live in one’s own house; “LCIF”: care facilities of the LTCI for people who require long-term care, such as a special nursing home; “SNH”: small-scale special nursing home or group home for people with dementia; “LWF”: welfare facilities for low-income older adults, such as low- cost nursing homes; “FSH”: fee-based nursing home/nursing home with diverse services and high costs; “OTHS”: other situations. “OR”: odds ratio. “SE”: standard error. ***: *p* < 0.01, **: *p* < 0.05, *: *p* < 0.1.

**Table 3 healthcare-11-01843-t003:** Marginal effects of household structure and economic status and their interaction term on preference for care facilities estimated from probit models.

**Variables**	**LIH**			**LCIF**			**SNH**		
	**ME**		**SE**	**ME**		**SE**	**ME**		**SE**
Household structure (ref: other categories)									
Living alone	−0.128	***	0.022	0.020	**	0.010	0.001		0.008
Economic status (self-assessed) (ref: other categories)									
Experiencing poverty	−0.079	***	0.016	0.002		0.006	0.007		0.006
Interaction term									
Living Alone and Experiencing Poverty	0.033		0.028	0.003		0.012	−0.013		0.009
Sex (ref: female)									
Male	0.048	***	0.013	−0.000		0.005	−0.003		0.005
Age (ref: 80+ y)									
65–69 y	−0.036	*	0.020	−0.004		0.008	0.004		0.008
70–74 y	−0.020		0.019	0.005		0.008	0.001		0.007
75–79 y	−0.021		0.019	0.009		0.008	−0.004		0.007
Health status									
IADL (dependent)	0.017		0.021	−0.004		0.008	−0.002		0.008
Subjective health (healthy)	0.097	***	0.017	−0.016	**	0.008	−0.023	***	0.008
Number of obs	5036			5036			5036		
**Variables**	**LWF**			**FSH**			**OTHS**		
	**ME**		**SE**	**ME**		**SE**	**ME**		**SE**
Household structure (ref: other categories)									
Living alone	0.059	***	0.016	0.038	***	0.011	−0.005		0.007
Economic status (self-assessed) (ref: other categories)									
Experiencing poverty	0.091	***	0.011	−0.030	***	0.007	−0.007		0.005
Interaction term									
Living Alone and Experiencing Poverty	−0.027	**	0.013	−0.019	*	0.011	0.034	*	0.020
Sex (ref: female)									
Male	−0.030	***	0.008	−0.007		0.006	0.001		0.005
Age (ref: 80+ y)									
65–69 y	0.010		0.012	0.010		0.010	0.026	**	0.010
70–74 y	0.010		0.012	−0.007		0.008	0.021	**	0.009
75–79 y	0.004		0.011	0.004		0.009	0.008		0.008
Health status									
IADL (dependent)	−0.003		0.013	−0.001		0.010	−0.013	**	0.006
Subjective health (healthy)	−0.031	***	0.011	0.000		0.008	−0.010		0.007
Number of obs	5036			5036			5036		

Notes: “LIH”: live in one’s own house; “LCIF”: care facilities of LTCI for people who require long-term care, such as special nursing homes; “SNH”: Small-scale special nursing homes or group homes for people with dementia; “LWF”: welfare facilities for low-income older adults, such as low-cost nursing homes; “FSH”: fee-based nursing homes/nursing homes with diverse services and high costs; “OTHS”: other situations. “ME”: marginal effects. “SE”: standard error. ***: *p* < 0.01, **: *p* < 0.05, *: *p* < 0.1.

## Data Availability

It is possible to obtain the data by applying for information disclosure with the Longevity Support Division of the Longevity Society Department, Health and Welfare Bureau of Sakai City, Osaka Prefecture. Access link: https://www.city.sakai.lg.jp/shisei/gyosei/shishin/fukushi/kourei-kaigo_keikaku/75402820211117160542577.html (In Japanese) (accessed on 16 May 2023).

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
