# Peer review of "Empirical Analysis of Preferences of Older Adults for Care Facilities in Japan: Focusing on Household Structure and Economic Status"

_healthcare, 2023, doi:10.3390/healthcare11131843_

Round 1

Reviewer 1 Report

An Empirical Analysis of Preferences of the Older Adults for  Care Facilities in Japan: Focusing on Household Structure and  Economic Status.

This is an interesting study; however, the work will benefit from major modification to enhance the quality and contributions before it can be published.

1.     The abstract is too long, authors should summarize the key aspect and make it more concise. Particularly, the background information could be shortened.

2.     The background information in the abstract does not properly connect the research gap. This should be improved. This study is about preferences of elder care facilities, background information should be focused on this.

3.     Introduction: I suggest some of the figures should be moved to appendices. Can present the statistics without throwing in every table and figures.

4.     Methods =Measures: Authors should be transparent on how some of the variables were measured. E.g., how economic status/ poor/ wealthy was measured?

5.     Discussion: the discussion is too patchy. What do the findings mean for policy and practice? How do the findings add to existing literature and or differ from earlier reports around the world? Authors should consider expanding the discussion section to address these issues. The current discussion falls short of a full-length research article.

6.     Conclusion is poorly written, and this may be due to the poor nature of the discussion section. Conclusion should capture take-home message/ key findings and their policy and practice implications. This, I cant find here.

Good luck!

acceptable

Reviewer 2 Report

The research topic this article addresses is quite relevant, although the authors do not necessarily make that relevance explicit.  Understanding preferences and expectations in a fast aging society is essential to plan future needs for formal care. Looking at choices considering variations in income and household models is rather intuitive and one would say not new. Controlling for both and discussing their relative weight is where the paper's novelty is since this will help services brace for changes. One would expect some more literature review about preferences and their predictors and not so much data on how Japanese society is aging fast.

Choices of data are clearly presented and the authors describe well data sources and characteristics.

Data analysis raises some questions. Logistic regression models were used but authors present and read probabilities. You should be presenting the odds ratios and not write about probablities. The marginal effects in the models are the b coefficients? If yes, they are not changes in probabilities. They are changes in logits. Authors report r squared which is pretty meaningless for a multinomial logistic regression model. Shouldn't you be reporting other goodness-o-fit measures? Tables need to undergo verification for statistical soundness.

The discussion section does not offer any discussion but rather repeats the findings, already discussed in the section on findings. It would be expected to have authors drawing some conclusions about expressed preferences and their predictors in what concerns the literature on the topic and the practical implications for policy development in Japan.

Once article is revised it would benefit from some proof-reading by a native speaker or a professional editor.

Round 2

Reviewer 2 Report

The authors have addressed all issues raised in the first review and have submitted a revised version that is now adequate for publication.

It is always good to have the final version revised by a native. I am not fully qualified to assess the soundness of the English but my impression is that it can benefit from a final revision.